# Genome-Wide Identification and Characterization of WRKY Transcription Factors and Their Expression Profile in *Loropetalum chinense* var. *rubrum*

**DOI:** 10.3390/plants12112131

**Published:** 2023-05-27

**Authors:** Yang Liu, Yifan Zhang, Yang Liu, Ling Lin, Xingyao Xiong, Donglin Zhang, Sha Li, Xiaoying Yu, Yanlin Li

**Affiliations:** 1College of Horticulture, Engineering Research Center for Horticultural Crop Germplasm Creation and New Variety Breeding (Ministry of Education), Hunan Mid-Subtropical Quality Plant Breeding and Utilization Engineering Technology Research Center, Hunan Agricultural University, Changsha 410128, China; 418272846@stu.hunau.edu.cn (Y.L.); zyfan@stu.hunau.edu.cn (Y.Z.); liuyang1203@stu.hunau.edu.cn (Y.L.); xingxingyao@caas.cn (X.X.); donglin@uga.edu (D.Z.); 2School of Economics, Hunan Agricultural University, Changsha 410128, China; llhnfx@126.com; 3Agricultural Genomics Institute at Shenzhen, Chinese Academy of Agricultural Sciences, Shenzhen 518120, China; 4Kunpeng Institute of Modern Agriculture, Foshan 528225, China; 5Department of Horticulture, University of Georgia, Athens, GA 30602, USA; 6College of Chemistry and Materials Engineering, Zhejiang A & F University, Hangzhou311300, China; shali@zafu.edu.cn; 7School of Biological Sciences, Nanyang Technological University, 60 Nanyang Drive, Singapore 637551, Singapore

**Keywords:** WRKY, *Loropetalum chinense* var. *rubrum*, genome-wide analysis, expression pattern, light quality

## Abstract

The *WRKY* gene family plays important roles in plant growth and development, as well as in the responses to biotic and abiotic stresses. *Loropetalum chinense* var. *rubrum* has high ornamental and medicinal value. However, few *WRKY* genes have been reported in this plant, and their functions remain unknown. To explore the roles that the *WRKY* genes play in *L. chinense* var. *rubrum*, we identified and characterized 79 *LcWRKYs* through BLAST homology analysis and renamed them (as *LcWRKY1–79*) based on their distribution on the chromosomes of *L. chinense* var. *rubrum*. In this way, according to their structural characteristics and phylogenetic analysis, they were divided into three groups containing 16 (Group I), 52 (Group II), and 11 (Group III) WRKYs, respectively. LcWRKYs in the same group have similar motifs and gene structures; for instance, Motifs 1, 2, 3, 4, and 10 constitute the WRKY domain and zinc-finger structure. The *LcWRKY* promoter region contains light response elements (ACE, G-box), stress response elements (TC-rich repeats), hormone response elements (TATC-box, TCA-element), and MYB binding sites (MBS, MBSI). Synteny analysis of *LcWRKYs* allowed us to establish orthologous relationships among the *WRKY* gene families of *Arabidopsis thaliana*, *Oryza sativa*, *Solanum lycopersicum* L., *Vitis vinifera* L., *Oryza sativa* L., and *Zea mays* L.; furthermore, analysis of the transcriptomes of mature leaves and flowers from different cultivars demonstrated the cultivar-specific *LcWRKY* gene expression. The expression levels of certain *LcWRKY* genes also presented responsive changes from young to mature leaves, based on an analysis of the transcriptome in leaves at different developmental stages. White light treatment led to a significant decrease in the expression of *LcWRKY6, 18, 24, 34, 36, 44, 48, 61, 62,* and *77* and a significant increase in the expression of *LcWRKY41*, blue light treatment led to a significant decrease in the expression of *LcWRKY18, 34, 50,* and *77* and a significant increase in the expression of *LcWRKY36* and *48*. These results enable a better understanding of *LcWRKYs*, facilitating the further exploration of their genetic functions and the molecular breeding of *L. chinense* var. *rubrum*.

## 1. Introduction

The WRKY transcription factor family is one of the largest families of transcription factors in plants [1]. One of the characteristics of this gene family is that it has a conserved WRKY domain—a 60-amino-acid region that is highly conserved amongst family members [2]—constituting DNA binding regions capable of specifically binding the W-box elements of target genes and, thus, transcribing the expression of other genes [3]. *SPF1*, the first of the WRKY cDNAs, was cloned from sweet potato [4]. Subsequently, corresponding *WRKY* genes were cloned in *Avena fatua* [5], *Petroselinum crispum* [6], and *Arabidopsis* [7]. With the development of gene sequencing technology [8], the *WRKY* gene family has been identified and characterized in the genomes and transcriptomes of an increasing number of species; for example, 72 and 82 *WRKY* genes have been found in the genomes of the dicotyledonous plant *Arabidopsis thaliana* [9] and tomato [10], respectively. Furthermore, the *WRKY* gene family has been identified in monocot plants, including rice [9] and maize [11]. The *WRKY* gene family is divided into three major groups, based on the WRKY domains and specific features of the zinc-finger-like motifs [2,12,13]. Group I contains two WRKY domains, Group II contains one WRKY domain and zinc-finger motifs of the C2H2 type, and Group III contains one WRKY domain and zinc-finger motifs of the C2HC type.

In plants, the *WRKY* genes have been carefully studied and shown to be extensively involved in growth and development [14], as well as in the responses to biotic stresses [15] and abiotic stresses [16]. Demonstrating the participation of WRKYs in plant growth and development, Jiani-Chen reported that *AtWRKY46*, *54*, and *70* are involved in Brassinosteroid (BR)-regulated growth as important signaling components [17]. In rice, *OsWRKY55* reduced the plant height in rice by decreasing the cell size [18], and the over-expression of *OsWRKY21* resulted in a rice semi-dwarf phenotype [19]; these results have important implications for research on rice’s resistance to lodging. Some *WRKY* genes have been reported to be involved in plant biotic stress responses when plants are affected by specific diseases with greater impact. For instance, *OsWRKY31* is induced by the rice blast fungus *Magnaporthe grisea*, and the overexpression of *OsWRKY* enhances resistance against *M. grisea* infection in rice [20]. In contrast, overexpression of *OsWRKY62* weakens the expression of the defensive gene and *Xa21*-mediated resistance to *Xanthomonas oryzae* pv. *Oryzae* (*Xoo*) [21]. Almost 57% (i.e., 16 genes) of the *WRKY* genes in *Vitis vinifera* were differentially regulated in response to pathogen infection [22]. Plants may be affected by various abiotic stresses during growth and development, such as salt stress, cold stress, heat stress, drought, and so on. Under salt stress, the *AhWRKY75* gene enhanced the efficiency of the ROS scavenging system in transgenic peanut and conferred salt tolerance [23]. *KoWRKY40* enhanced cold tolerance in transgenic *Arabidopsis* [24]. *ZmWRKY106* improved the tolerance to drought and heat in transgenic *Arabidopsis* [25]. In summary, increasing reports have demonstrated that the *WRKY* genes occupy an important position in plants; therefore, it is considered worthwhile to identify the *WRKY* genes in various plant species.

Light plays a key role in the growth and development of plants [26], modulating the biosynthesis of specialized metabolites [27]. Eva Darko et al. [28] published a report describing how artificial lighting can provide plants with the energy and information required for their development, including the effects of different light quality levels on plants [27]. A red–blue LED incubator led to the better growth of lentils and basil, as well as a higher number of flower buds and fewer days to flowering for pot flowers [29]. Ultraviolet-A-specific induction of anthocyanin biosynthesis has been reported in *Brassica rapa* [30]. The *WRKY* genes have also recently been shown to play a role in photomorphogenesis [31]. ELONGATED HYPOCOTYL5 (HY5), which belongs to the bZIP transcription factor family [32], and CONSTITUTIVELY PHOTOMORPHOGENIC 1 (COP1) are central components of photomorphogenesis [31]. Hua Zhou et al. [33] reported that *WRKY32* activates *HY5* transcription through binding to its promoter and promotes the development of photomorphogenesis through the COP–HY5 signaling pathway. *WRKY36* binds to the W-box motif of the promoter of *HY5* to inhibit its transcription. In contrast, UV RESISTANCE LOCUS 8 (UVR8) interacts with *WRKY36* to inhibit *WRKY36*–DNA binding in vitro and in vivo, thereby attenuating the transcriptional repression of *HY5* [34]. These results indicate that *WRKY36* is involved in the photomorphogenesis regulated by *HY5* and interacts with *UVR8*. The involvement of the *WRKY* genes in the light response in plants still requires a significant amount of research.

*L. chinense* var. *rubrum* is a variety of *Loropetalum chinense* (R. Br.) Oliver in the family Hamamelidaceae, which has both ornamental and medicinal value. The flowers are red when in bloom, the tree is beautiful, and its ornamental value is high, and it has the characteristics of strong ecological adaptability, easy reproduction, resistance to pruning, easy shaping, and so on. As such, it is referred to as an “all-around variety” in the garden industry. The *WRKY* gene family plays a very important role in plants. However, few studies have focused on the *WRKY* genes in *L. chinense* var. *rubrum.* In this study, the *WRKY* gene family is identified and characterized for the first time, through whole-genome analysis of *L. chinense* var. *rubrum.* The physicochemical properties, phylogenetic development, gene structure, gene duplication, and synteny relationships with other species regarding the *LcWRKY* genes are described. Furthermore, the expression patterns of the *LcWRKY* genes in different transcriptomes are described, with the aim of discovering interesting aspects. Our study provides theoretical support for the further elucidation of the functions of the *WRKY* genes in *L. chinense* var. *rubrum*, facilitating the molecular breeding of new varieties.

## 2. Results

### 2.1. Identification and Characterization of LcWRKYs

In the genome of *L. chinense* var. *rubrum*, 79 WRKY-encoding genes containing the complete WRKY protein domain were identified through two methods. The protein sequences of 79 LcWRKYs contained the complete WRKY domain (Figure 1); the UPF0242 protein domain (Pfam ID: PF06785) was found in the protein sequences of LcWRKY3, 2, and 5; and LcWRKY76, 20, 7, 6, 55, 53, and 54 contained the Plant_zn_clust protein domain (Pfam ID: PF10533). Meanwhile, an incomplete bZIP protein domain was observed in the protein sequences of LcWRKY50 and 79. The NCBI gene accession numbers of the 79 *LcWRKY* genes are provided in Appendix A.

The 79 *LcWRKY* genes were renamed (as *LcWRKY1–LcWRKY79*) according to their locations on the chromosomes. The 79 *LcWRKY* genes were found to be unevenly distributed on the 12 chromosomes (Figure 2), with chromosome 2 possessing the fewest *WRKY* genes (3 *LcWRKY* genes) and chromosomes 3 and 11 having the highest number of *WRKY* genes (10 *LcWRKY* genes). The number of *WRKY* genes on chromosomes 1, 4, 5, 6, 7, 8, 9, 10, and 12 were 7, 7, 6, 6, 5, 8, 4, 6, and 7, respectively.

The *LcWRKY* genes’ isoelectric point (pI) values ranged from 4.84 (*LcWRKY4*) to 9.75 (*LcWRKY72*). The smallest molecular weight (Mw) was 15,228.96 Da (*LcWRKY42*), while the largest was 87,410.2 Da (*LcWRKY56*). The length of the proteins encoded by the *LcWRKY* genes (aa length) ranged from 131 (*LcWRKY41*) to 794 (*LcWRKY56*). Additional protein features of the *LcWRKY* genes are shown in Appendix A, and the sub-cellular localization of all LcWRKY proteins was predicted to be nuclear. The differences in the characteristics of LcWRKY proteins imply that they each have different functions in different microenvironmental contexts.

### 2.2. Phylogenetic Analysis of the LcWRKYs, Classification, and Multiple Sequence Alignment

The WRKY structural domain of WRKY transcripts is highly conserved and is equally conserved across species [35]. To explore the evolutionary relationships among LcWRKYs, we established a phylogenetic tree containing 72 AtWRKY and 79 LcWRKY proteins, as shown in Figure 3. Based on the multiple sequence alignment of the full lengths of the LcWRKYs (Figure 4) and the phylogenetic tree, the 79 WRKYs of *L. chinense* var. *rubrum* could be divided into three groups [2], namely Groups I, II, and III, with 16, 52, and 11 members, respectively. Group II was divided into five sub-groups, based on the primary amino acid sequences [12], where sub-groups IIa, Iib, Iic, Iid, and Iie had 4, 13, 20, 8, and 7 members, respectively. The phylogenetic tree results indicated that LcWRKY18 and AtWRKY51 in Group Iic, LcWRKY28 and AtWRKY9 in Group Iib, and LcWRKY31 and AtWRKY27 in Group Iie belonged to the same clusters.

According to the classification of WRKYTFs, those containing two WRKY structural domains belong to Group Ⅰ [35]. We found three specific LcWRKYs in Group I in the multiple sequence comparison results for the 79 full-length LcWRKY amino acid sequences (Figure 4). For example, in LcWRKY68 and 77, the C-terminal protein sequence contains only the zinc-finger structure of C2H2, while the LcWRKY14 C-terminal WRKY domain is incomplete. Based on the positions of these three LcWRKYs on the phylogenetic tree (Figure 3), we arbitrarily categorized them as Group I. The sequence conserved at the N-terminal end in the WRKY transcription factor family was WRKYGQK [2]. Four sequences different from WRKYGQK—namely, WRKYGKK (LcWRKY18, 22, 23), WRKYGEK (LcWRKY29), WRKYGEK (LcWRKY 30), and WRKYAET (LcWRKY 32)—were identified in the LcWKRYs. These WRKY structural domain changes have also been observed in other species [36]. Notably, WRKYGQK remained highly conserved in the 79 LcWRKY proteins.

### 2.3. Analysis of Conserved Motifs and Gene Structures

To understand the relationships between the LcWRKY proteins, the conserved motifs in the 79 LcWRKYs and the structures of the *LcWRKY* genes were analyzed. Using the MEME program, we searched for conserved motifs (we set the number to 10) in the protein sequences of the 79 LcWRKYs. The sequence widths of the searched motifs ranged from 21 to 50. Additionally, motifs 1, 2, 3, 4, and 10 constitute the WRKY domain and zinc-finger structure (Appendix A). From Figure 5, it can be seen that LcWRKY proteins belonging to the same family or sub-family have similar conserved motifs; for example, Group I members contain motifs 3, 10, 2, and 4, with motifs 3 and 10 only in Group I and Group IIc, while only Group IIa and Group IIb contain motifs 6 and 9. The exon/intron patterns of the *LcWRKY* genes were diverse (Figure 5). The distribution of the number of introns in the *LcWRKY* genes was 1–7, most of the *LcWRKY* genes contained two introns (37/46.83%), and *LcWRKY37* had the largest number of introns (with seven). The numbers of *LcWRKY* genes containing 1, 3, 4, 5, and 6 introns were 6, 7, 12, 12, and 4, respectively. The number of exons ranged from two to eight, and 46.83% (37) of the *LcWRKY* genes contained two exons, which was the highest number. As with introns, the gene possessing the highest number of exons was also the *LcWRKY37* gene (with eight exons). In addition, the numbers of genes possessing 2, 4, 5, 6, and 7 exons were 6, 7, 11, 13, and 4, respectively. The different exon/intron patterns of the *LcWRKY* genes may suggest that they present functional differences.

### 2.4. cis-Element Analysis of LcWRKY Genes

The upstream 2000-bp sequences with respect to the *LcWRKY* genes were extracted for *cis*-acting element prediction. Details of the *cis*-acting elements associated with the 79 predicted *LcWRKY* genes obtained are provided in Appendix A, and some of the components of interest are detailed in Figure 6. 

It is obvious, from the results that light-responsive elements were present in the promoters of all *LcWRKY* genes. Stress response elements included TC-rich repeats, LTR, and the WUN-motif. The Tatc-box, TCA-element, ABRE, TGACG-motif, CGTCA-motif, P-box, and Care-motif were included among the hormone-responsive elements. MYB binding sites (MBS, MBSI, MRE, CCAAT-box) were also observed. In particular, only the promoter region of *LcWRKY62* contained a *cis*-acting element involved in phytochrome downregulation among all *LcWRKY* genes. The promoter region of *LcWRKY2* had the most *cis*-acting elements (40 sites). The promoter regions of the 79 *LcWRKY* genes contained elements that might be related to hormones and stress, suggesting that the *LcWRKY* genes are involved in biotic and abiotic stress response mechanisms.

### 2.5. Duplication and Synteny Analysis of LcWRKY Genes

We investigated gene duplication events in the LcWRKY genes, and 32 LcWRKY gene pairs were found between the 12 chromosomes of *L. chinense* var. *rubrum*. Figure 7A shows five tandem duplicated gene pairs (LcWRKY2–LcWRKY3, LcWRKY32–LcWRKY33, LcWRKY40–LcWRKY41, LcWRKY71–LcWRKY72, and LcWRKY8–LcWRKY9) on chromosomes 1, 2, 5, 7, and 11, respectively. A total of 36 segment duplication pairs were identified in the *L. chinense* var. *rubrum* chromosomes. By analyzing the Ka/Ks values of these 32 LcWRKY gene pairs, except for two pairs of LcWRKY genes (LcWRKY51–LcWRKY52 and LcWRKY6–LcWRKY7), it was found that the Ka/Ks values were all less than 1, indicating that these LcWRKY genes have undergone purifying selective pressure (Appendix A).

To explore the evolutionary mechanism of the WRKY family in *L. chinense* var. *rubrum*, comparative syntenic maps regarding *L. chinense* var. *rubrum* and five other species were constructed, including three dicotyledonous plants (*Arabidopsis thaliana*, *Solanum lycopersicum*, and *Vitis vinifera* L.) and two monocotyledons (*Oryza sativa* and *Zea mays* L.). A total of 50 (*Arabidopsis thaliana*), 78 (*Solanum lycopersicum* L.), 87 (*Vitis vinifera* L.), 23 (*Oryza sativa* L.), and 16 (*Zea mays* L.) pairs of homologous *WRKY* genes were found in these five different species (Figure 7B). Some *LcWRKY* genes had multiple homologous genes in the other five species; for example, *LcWRKY6* had three and four homologous genes in maize and rice, respectively. *LcWRKY35* had the most (12) homologous genes among the five species, which implies that *LcWRKY35* may have played an important role in the evolution of the WRKY family. Overall, we found a total of 58 *LcWRKY* genes with homologous genes (Appendix A). 

### 2.6. Expression Analysis of the LcWRKY Genes in Leaves and Flowers of Different Cultivars

The expression of the *LcWRKY* genes varied between the leaves of different varieties; these differences are shown in the heat map in Figure 8A. Interestingly, the transcripts of *LcWRKY78* were not detected in the four considered varieties (‘XNXY’, ‘XNXJ’, ‘XNFJ’, and ‘XNNC’), while other WRKY genes were read in the transcripts of all four varieties (FPKM > 0). *LcWRKY46*, *LcWRKY8*, and *LcWRKY47* were only highly expressed in ‘XNNC’, while 10 *LcWRKY* genes (*LcWRKY35, 34, 20, 32, 31, 26,17, 3, 27,* and *65*) were highly expressed only in ‘XNFJ’. *LcWRKY55, 76, 6,* and *77* had higher expression in ‘XLXJ’, while no *LcWRKY* genes were highly expressed only in ‘XNXY’. There were clear differences in the transcriptome data of the four varieties regarding the *LcWRKY* genes, which may be related to the different leaf phenotypes of the four varieties. The *LcWRKY* expression patterns in flowers of different varieties are shown in Figure 8B. It is worth noting that 49 *LcWRKY* genes were highly expressed in ‘XNNC’, accounting for 62% of all *LcWRKY* genes. Eight of these genes were only highly expressed in ‘XNNC’ (*LcWRKY52, 8, 28, 70, 47, 6,* and *49*). *LcWRKY59, 61, 7, 38, 39, 73, 1, 40, 36, 16, 44, 62, 74, 77, 24, 35,* and *43* were highly expressed in the flowers of ‘XNFJ’ and ‘XNNC’ and were low in the flowers of ‘XNXY’ and ‘HYJM1′. These differentially expressed *LcWRKY* genes may have an important relationship with the phenotype, and *LcWRKY* genes with certain functions can be subsequently mined.

### 2.7. Expression Analysis of the LcWRKY Genes in Leaves at Different Developmental Stages

Next, we explored the *LcWRKYs* that could potentially play a role in leaf development in *L. chinense* var. *rubrum*. Figure 8C shows an expression heat map for the *LcWRKYs* in leaves at different developmental stages in the four varieties: young leaves (I) and mature leaves (II). The expression levels of certain *LcWRKY* genes showed significant trends at different stages of development. The changes in *LcWRKY* gene expression between the groups ‘XNXY.I’ and ‘XNXY.II’ were stable, except for three genes (*LcWRKY5, LcWRKY58,* and *LcWRKY75*). In ‘XNFJ.I’ and ‘XNFJ.II’, the expression levels of most of the *LcWRKY* genes were stable between the two stages. *LcWRKY* genes with highly reduced expression between ‘XNNC.I’ and XNNC.II’ included *LcWRKY9, 25, 31, 37, 39, 40, 55, 57, 66, 70, 75,* and *78,* while *LcWRKY58* presented an elevated expression level. Between ‘XNXJ.I’ and ‘XNXJ.II’, 15 *LcWRKYs* presented increased expression (the most among the four species), while those with decreased expression included *LcWRKY56, 42, 57, 37, 51, 55, 12, 16, 39, 45, 52,* and *15.* In summary, the expression of *LcWRKY* genes varied between different developmental stages of the leaves, and it can be concluded that *LcWRKYs* may be related to the growth and development of *L. chinense* var. *rubrum.*

### 2.8. The Expression Analysis of the LcWRKY Genes under White and Blue Light Treatment

Light has a significant impact on plant growth and development, as well as the synthesis of secondary metabolites [27,28,31,37]. Based on the analysis of *cis*-acting elements 2000 bp upstream of the *LcWRKY* genes, we found that the promoter regions of all *LcWRKY* genes contained light-responsive elements. Subsequently, we further analyzed the expression changes of the *LcWRKY* genes in *L. chinense* var. *rubrum* under the influence of light. Based on the results of the previous gene classification and *cis*-acting element predictions, we selected *LcWRKY* members distributed in different sub-groups, including *LcWRKY6*, *LcWRKY18*, *LcWRKY24*, *LcWRKY34*, *LcWRKY36*, *LcWRKY41*, *LcWRKY44*, *LcWRKY48*, *LcWRKY50*, *LcWRKY61*, *LcWRKY62*, and *LcWRKY77*, for RT-PCR analysis under white and blue light treatment. In Figure 9, he RT-PCR results indicated that the relative expression levels of *LcWRKY6, 18, 24, 34, 36, 44, 48, 50, 61, 62,* and *77* were lower after 5 days of white light treatment than at day 0. Among these, the relative expression levels of *LcWRKY6, 18, 24, 34, 48, 61, 62,* and *77* were extremely significantly lower on day 5 than on day 0 after white light treatment, while the relative expression of *LcWRKY44* was significantly lower than that on day 0. Only the relative expression of *LcWRKY41* was significantly higher than that on day 0. After 5 days of blue light treatment, the relative expression levels of *LcWRKY18, 34,* and *77* were significantly decreased compared to day 0, while the relative expression level of *LcWRKY50* was extremely significantly lower than on day 0. However, the relative expression levels of *LcWRKY36* and *48* were significantly higher than on day 0 after 5 days of blue light treatment. Overall, both white and blue light led to significant decreases in the relative expression of *LcWRKY18, 34,* and *77.*

## 3. Discussion

The *WRKY* gene family is one of the largest families of transcription factors in flowering plants [38]. Since the first *WRKY* gene was identified in sweet potato in 1994 [4], extensive studies have shown that *WRKY* genes play important roles in plant growth and development, as well as in the responses to biotic and abiotic stresses [12,16]. With the rapid development of sequencing technology, genome sequencing results for many species have recently been reported; for example, using bioinformatics techniques, the *WRKY* genes have been identified and characterized in different species [39,40], with 72, 102, and 119 *WRKY* genes having been found in Arabidopsis [9], rice [41], and maize [11], respectively.

*L. chinense* var. *rubrum* is a plant originating from Hunan, China, with colorful foliage and a wide range of garden uses, as well as medicinal value. To date, there have been no studies focused on mining the *WRKY* genes in *L. chinense* var. *rubrum*. With the completion of the sequencing of the *L. chinense* var. *rubrum* whole genome [42], we identified and characterized 79 *WRKY* genes. The current classification of *WRKY* genes is mainly based on the number of WRKY domains and the differences in the zinc-finger structure [2,12], and many species can be classified based on this approach regarding the *WRKY* genes. According to the analysis of the multiple sequence alignment and phylogenetic tree results, the 79 *LcWRKY* genes were mainly divided into three groups: Group Ⅰ containing 16 members (20%), Group II containing 52 members (66%; the most of the three groups), and Group III containing 11 members (14%).

We found that Group Ⅰ contained three unique LcWRKY members. In general, Group I members of the WRKY gene family contain two WRKY domains [2], while three LcWRKYs (LcWRKY14, 68, and 77) contain only one complete WRKY domain (Figure 4**)**. We considered removing these three LcWRKYs from Group I but, based on their branching positions in the phylogenetic tree, we classified these three LcWRKYs as Group Ⅰ. These differences may have been caused by the expansion of the gene family, a normal phenomenon [11,43]. Group Ⅱ was further divided into five sub-groups (IIa, IIb, IIc, IId, and IIe). The largest of the three groups was Group II, which may have undergone more gene duplication during the evolutionary process. The results of this classification were similar to those in other species, such as grapevine (*Vitis vinifera* L.) [44], *Isatis indigotica* [45], *Ophiorrhiza pumila* [46], *Solanum lycopersicum* [10], and *Eucommia ulmoides* [47]. Meanwhile, existing studies have suggested that WRKY Group II members play an important role in the responses to abiotic stresses [48]. ZmWRKY17 (in IId) negatively regulated salt stress tolerance in transgenic *Arabidopsis* plants [49]. As *AhWRKY75*-overexpressing peanuts grew better than wild-type peanuts after salt stress, salt tolerance in transgenic peanut lines was considered to be conferred by the *AhWRKY75* (IIc) gene [23]. GbWRKY1 (IIc) negatively regulated salt and drought tolerance through the ABA signaling pathway by participating in the interactive network of JAZ1 and ABI1 [50]. These and similar results imply that Group II LcWRKYs are also involved in the responses to abiotic stresses.

From the conserved motif analysis (Figure 5), the Group I members all contained motifs 3, 10, 5, 1, 2, and 4, except for the three special LcWRKYs (68, 77, and 14). The motifs contained in LcWRKYs within the same group were similar. For example, motifs 6, 7, and 9 were unique to Groups IIa and IIb. All LcWRKYs, except for LcWRKY 36, 43, 44, and 66, contained motifs 1 and 2. Motifs 1 and 2 and motifs 3 and 4 form a conserved WRKY domain within the WRKY gene family (Figure 5 and Appendix A). Although WRKYs have the same conserved WRKY domain, their overall structures have varying characteristics and they can be divided into different groups, indicating that they have different functions [2]. The gene structures within the same WRKY groups were also similar. LcWRKY Group III contained two introns, except LcWRKY21 (with 32). A similar situation has been observed in other species; for example, PhWRKY Group III also contains two introns [51]. Various types of *cis*-regulatory elements were identified in the upstream promoter regions of LcWRKYs (Figure 6), including TC-rich repeats, LTR, the WUN-motif, the CGTCA-motif, and so on. The analysis of the *cis*-elements indicated that *LcWRKYs* play a role in the response to various environmental stresses and affect hormone responsiveness.

Gene replication events play an important role in the expansion and evolution of gene families [52]. In total, we found five pairs of tandem duplication genes and thirty-four segment duplication pairs on the chromosomes of *L. chinense* var. *rubrum.* It can be considered that segment duplication was the main event in the expansion and evolution of the *LcWRKY* genes, consistent with results previously obtained in oat (*Avena sativa* L.) [53]. The thirty-four segment duplication pairs were subjected to strong purifying selection during evolution. Analysis of covariance in other species has shown that the dicots *Arabidopsis thaliana*, tomato, and grape have more homologous *WRKY* genes than monocots such as rice and maize (Figure 7B). Interestingly, the number of genes homologous to *LcWRKY* genes among dicotyledonous plants (*Arabidopsis thaliana*, *Solanum lycopersicum* L., *Vitis vinifera* L.) is typically larger than that in monocotyledonous plants (*Oryza sativa* L., *Zea mays* L.). This indicates that the closer the relationship between species, the greater the homology of the *WRKY* gene family. It can be conjectured that this is due to differences in plant expansion and evolution caused by the WRKY family of transcription factors.

Transcriptome data are often used for gene function mining [54,55]. To explore the potential functions of the *LcWRKY* genes, three different types of transcriptomic data were analyzed, including those from leaves and flowers of different varieties, leaves at different developmental stages within different varieties, and leaves under different light treatments. Certain *LcWRKYs* were specifically highly expressed in mature leaves of different variants, with very clear varietal differences. *LcWRKY46, 8,* and *47* were more highly expressed in ‘XNNC’ compared to the other varieties; the expression levels of *LcWRKY55, 76, 6,* and *77* were higher in ‘XNXJ’ than in the other three varieties; 10 *LcWRKY* genes were expressed to a higher degree in ‘XNFJ’ than in other varieties; and eight *LcWRKYs* were expressed more significantly in ‘XNNC.F’ than other varieties. These results imply that certain LcWRKYs play a role in the morphological differences between different varieties, and the associated relationships need to be explored further. Many studies have proven that the *WRKY* gene family plays an important role in the growth and development of plants [14]; for example, *WRKYs* are involved in seed germination and seedling growth [56,57], flowering [58], and fruit ripening [59]. In ‘XNXY.I’ and ‘XNXY.II.’, only three *LcWRKY* genes (*LcWRKY5, 58,* and *79*) presented significant changes. In leaves at different stages of development, *LcWRKYs* were up- and downregulated to varying degrees, and this may play a role in the leaf growth of *L. chinense* var. *rubrum*.

The leaf color of *L. chinense* var. *rubrum* depends on a mixture of chlorophyll, carotenoids, and flavonoids. However, artificial LED light affects the secondary metabolites in plants [27]. In this report, we analyzed the expression levels of some *LcWRKY* genes under while and blue light treatment. Interestingly, both white light and blue light led to a decrease in the expression of some *LcWRKY* genes. Whether genes involved in certain photo response pathways inhibit their expression and further affect certain physiological programs in plants needs to be explored further regarding the functions of *LcWRKY* genes. The *WRKY* gene family has been studied in many plant species, and we have identified and characterized *WRKY* gene family members in the whole genome of *L chinense* var. *rubrum* for the first time. The results provide an important basis for further study of the functions of *LcWRKY* genes, as well as facilitating the development of new ideas for the molecular breeding of *L. chinense* var. *rubrum*.

## 4. Materials and Methods

### 4.1. Identification, Chromosome Localization, and Characteristic Analysis of LcWRKYs

The genome data of *Loropetalum chinense* var. *rubrum* were provided by the ‘*Loropetalum chinense* var. *rubrum* Research Team’ of the College of Horticulture of Hunan Agricultural University. Sequencing data used in this study are available upon request to the corresponding author. To identify the *WRKY* genes in *L. chinense* var. *rubrum*, two approaches were followed. First, 72 WRKY protein sequences of *Arabidopsis thaliana* were downloaded from TAIR (https://www.arabidopsis.org/ (accessed on 4 April 2022)) and used as query sequences to identify homologous WRKY 60 in the *L. chinense* var. *rubrum* genome using the TBtools software [60]. Subsequently, these homologous WRKY members of *L. chinense* var. *rubrum* were BLASTed using the NCBI Protein BLAST program (https://blast.ncbi.nlm.nih.gov/Blast.cgi (accessed on 4 April 2022)), in order to obtain potential *LcWRKY* genes. Next, the hidden Markov model file of the WRKY domain (Pfam: PF03106) was downloaded as a query from the Pfam database [61] (http://pfam.xfam.org/ (accessed on 4 April 2022)) and analyzed using HMMER (Ver3.0), with default settings and e-value < 0.001 [45], in order to identify possible WRKY family members of *L. chinense* var. *rubrum.* Duplicate lists were removed from the obtained results by the two methods, and the protein sequences of potential LcWRKYs were submitted to NCBI Batch CD-Search (https://www.ncbi.nlm.nih.gov/Structure/bwrpsb/bwrpsb.cgi (accessed on 4 April 2022)). SMART (https://smart.embl.de/ (accessed on 4 April 2022)) [62] was used to remove sequences without WRKY domains. *LcWRKY* genes with intact WRKY domains were then preserved for further analysis. The locations of all *LcWRKY* genes on the chromosomes were determined in light of the genome annotation file, with the visualization of chromosome position maps performed using the MG2C website [63] (http://mg2c.iask.in/mg2c_v2.1/ (accessed on 23 March 2022)). According to the gene locations, we then re-named the *LcWRKY* genes. The sequences of protein-confirmed LcWRKYs were uploaded to the ExPASy website (https://www.expasy.org/ (accessed on 14 April 2022)), in order to calculate the physicochemical properties of the proteins, package molecular weights (MW), and isoelectric points (pI). The sub-cellular localization of the LcWRKYs was predicted using the Cell-PLoc 2.0 website [64] (http://www.csbio.sjtu.edu.cn/bioinf/Cell-Ploc-2/ (accessed on 30 November 2022)). The sequences of all LcWRKY proteins were submitted to the Batch CD-Search tool of NCBI (https://www.ncbi.nlm.nih.gov/Structure/bwrpsb/bwrpsb.cgi (accessed on 14 December 2022)) for analysis.

### 4.2. Construction of Phylogenetic Trees and Multiple Sequence Alignment

A total of 72 *Arabidopsis thaliana* WRKY protein sequences were downloaded from TAIR (https://www.arabidopsis.org/ (accessed on 4 April 2022)), and all protein sequences of the *WRKY* gene family from *L. chinense* var. *rubrum* were aligned using ClustalW. Phylogenetic trees were constructed using MEGA 11 [65] with the neighbor-joining method and bootstrap analysis (1000 replicates) [66]. The NJ tree containing the WRKY protein sequences of *Arabidopsis thaliana* and *L. chinense* var. *rubrum* was visualized and enhanced using the Evolview [67] online tool (https://www.evolgenius.info/evolview-v2 (accessed on 20 April 2022)). According to the WRKY conserved domain features and the phylogenetic tree for the classification of *LcWRKY* genes, an ML tree was constructed that contained the protein sequences of all *LcWRKY* genes, using MEGA7 [68]. The protein sequences of the LcWRKYs were aligned using the DNAMAN software (Ver 9.0), and Microsoft Paint was utilized to enhance the sequence alignment image; in particular, the structural features of LcWRKYs were marked.

### 4.3. Conserved Motifs and Gene Structure Analysis

The conserved motifs of the LcWRKY proteins were determined using the MEME online program (https://meme-suite.org/meme/tools/meme (accessed on 20 April 2022)) with the maximum number of motifs (10), and the size distribution (zoops) was displayed using the TBtool software [60]. Based on the *L. chinense* var. *rubrum* genome, an annotation file displaying the structure of the *LcWRKY* genes (including introns and exons) was also obtained using the TBtools program [60].

### 4.4. cis-Regulatory Element Analysis

The sequences 2000 bp upstream of all the *LcWRKY* genes were extracted, using Tbtools [60], from the *L. chinense* var. *rubrum* genome database, and then uploaded to the PlantCARE database (http://bioinformatics.psb.ugent.be/webtools/plantcare/html/ (accessed on 4 August 2022)) for the analysis of potential *cis*-regulatory elements. We selected some of the *cis*-regulatory elements of interest from the predicted results and visualized them using the R package ggplot2 [69].

### 4.5. Gene Duplication, Calculation of (ka/ks) Non-Synonymous/Synonymous Ratio, and Synteny Analysis

The One-Step MCScanX-SuperFast program of Tbtools [60] was used to perform gene duplication pair and synteny analyses, with the results visualized using Advanced Circos of Tbtools [70]. Calculation of Ka/Ks values for homologous gene pairs between LcWRKYs was conducted using Tbtools [60]. The analysis of syntenic relationships between *L. chinense* var. *rubrum* and the other five species (three dicotyledonous plants, *Arabidopsis thaliana*, *Solanum lycopersicum*, and *Vitis vinifera* L.; and two monocotyledons, *Oryza sativa* and *Zea mays* L.) was conducted using One-Step MCScanX-SuperFast of Tbtools [60] and visualized using Dual Systeny Plot of Tbtools [60].

### 4.6. Generation of Transcriptome Data and the Gene Expression Pattern Analysis

Plant materials used for transcriptome sequencing included one *Loropetalum chinense* variant, ‘XNXY’ (‘Xiangnong Xiangyun’), and four *L. chinense* var. *rubrum* variants, ‘XNXJ’ (‘Xiangnong Xiaojiao’), ‘XNFJ’ (‘Xiangnong Fenjiao’), ‘XNNC’ (‘Xiangnong Nichang’), and ‘HYJM1′ (‘Huaye Jimu 1′), which were planted in the flower center at Hunan Agricultural University. The preparation of RNA samples of leaves and flowers and the library construction and sequencing referred to a previous study [71,72]. In brief, total RNA was extracted using TRIzol reagent (GenStar P124-01). The detection of RNA degradation was performed using 1% agarose gel and a NanoDrop (NanoDrop Products, Wilmington, NC, USA) to detect RNA contamination. Total RNA was assessed using the Qubito RNA AssaKit in the Oubit@2.0 Fluorometer (Life Technologies, South San Francisco, CA, USA) and an Agilent 2100 Bioanalyzer (Agilent Technologies, Santa Clara, CA, USA), respectively, to determine the quantity and quality of RNA. Afterward, 2 × 150 paired-end RNA-seq was performed using the ILLUMINA NovaSeq 6000. All qualified RNA samples were stored at −20 °C and used for SMRT sequencing and RNA-seq sequencing within one week. All raw data were quality-controlled by fast QC v0.11.2, including adapters, low readings, and ploy-N. In addition, using Bowtie2 (v2.2.5), clean data were obtained and mapped to the final transcripts of the full-length transcriptome of *L. chinense* var. *rubrum*. Estimation of the reads per transcript was performed via the expected fragment number per thousand transcript sequences per million sequenced base pairs (FPKM) [71,72]. The raw transcriptome data reported in this paper were deposited in the Genome Sequence Archive (Genomics, Proteomics & Bioinformatics 2021) at the National Genomics Data Center (Nucleic Acids Res 2022), as well the China National Center for Bioinformation of the Beijing Institute of Genomics, Chinese Academy of Sciences (GSA: CRA009284 and CRA009285), and they are publicly accessible at https://ngdc.cncb.ac.cn/gsa (accessed on 10 May 2023) [73]. All transcriptome data are expressed using FPKM (reads per kilobase of exon model per million mapped reads). All expression data for log_2_ processing, log_2_(FPKM+1), and visualization were treated using the Peatmap and Circlize [74] packages of R.

### 4.7. Plant Material and Treatments

The used plant material was from the *L. chinense* var. *rubrum* variety ‘Hei Zhenzhu’, which is commonly used in landscaping and yields triennial seedlings. The cultivation substrate was composed of pastoral soil, vermiculite, and perlite in a 3:1:1 ratio. The white and blue light quality settings were as follows: light quality source selection, custom LED lamp; white light, 390–780 nm; blue light, 460 ± 5 nm; and light intensity, approximately 200 μmol/m^2^/s. The light source was installed at a height of 15 cm above the plant. A total of 30 pots of plant material were used per treatment. The photoperiod was 14 h/10 h (day/night), with a temperature of 24 °C/20 °C (day/night) and humidity of 65–75%. The experiment was conducted at the flower center of Hunan Agricultural University.

### 4.8. Real-Time Fluorescence Quantitative PCR

Total RNA was extracted from plant samples using a SteadPure Plant RNA Extraction Kit (Accurate Biotechnology, Hunan), according to the manufacturer’s instructions. cDNA was synthesized in 500 ng total RNA using an Evo M-MLV reverse transcription kit (Accurate Biotechnology, Hunan). Primer design of genes was carried out using the online website https://www.genscript.com/tools/real-time-pcr-taqman-primer-design-tool (accessed on 9 May 2023) (see Appendix A). The system and procedure of RT-PCR referred to Zhang et al. [75]. In brief, the total RT-PCR system amounted to 10 µL, including 5 µL of 2X SYBR Green Pro Taq HS Premix*, 1 µL of cDNA, 0.8 µL each of upstream and downstream primers (10 µmol/L), and ddH_2_O. The RT-PCR reaction procedure was as follows: step 1, 95 °C for 30 s; step 2, 95 °C for 5 s, 60 °C for 30 s, 72 °C for 10 min for 40 cycles, 65 °C for 5 s, 95 °C for 5 s; 3 repetitions. The relative expression of genes was calculated by the 2^−ΔΔCt^ method.

## 5. Conclusions

In the whole genome of *L. chinense* var. *rubrum*, we identified and characterized 79 *LcWRKY* genes distributed on 12 chromosomes. We then explored the gene structure of *LcWRKY*, the phylogenetic relationships, and the expression patterns with respect to different transcriptomes. Among the obtained genes, according to the classification based on the WRKY domain and zinc-finger structure, Group Ⅰ had 16 members, Group II had 52 members, and Group III had 11 members. The expression pattern results indicated that the expression levels of *LcWRKYs* significantly differed among different ecotypes. Moreover, the expression levels of some *LcWRKYs* were differentially affected under white and blue light treatments, including *LcWRKY6, 18, 24, 36, 48,* and *50.* These results can be used as a basis for further in-depth study. Overall, this article provides a preliminary analysis of the *LcWRKY* gene family, facilitating further exploration of the molecular function of *LcWRKY* genes.

## Figures and Tables

**Figure 1 plants-12-02131-f001:**
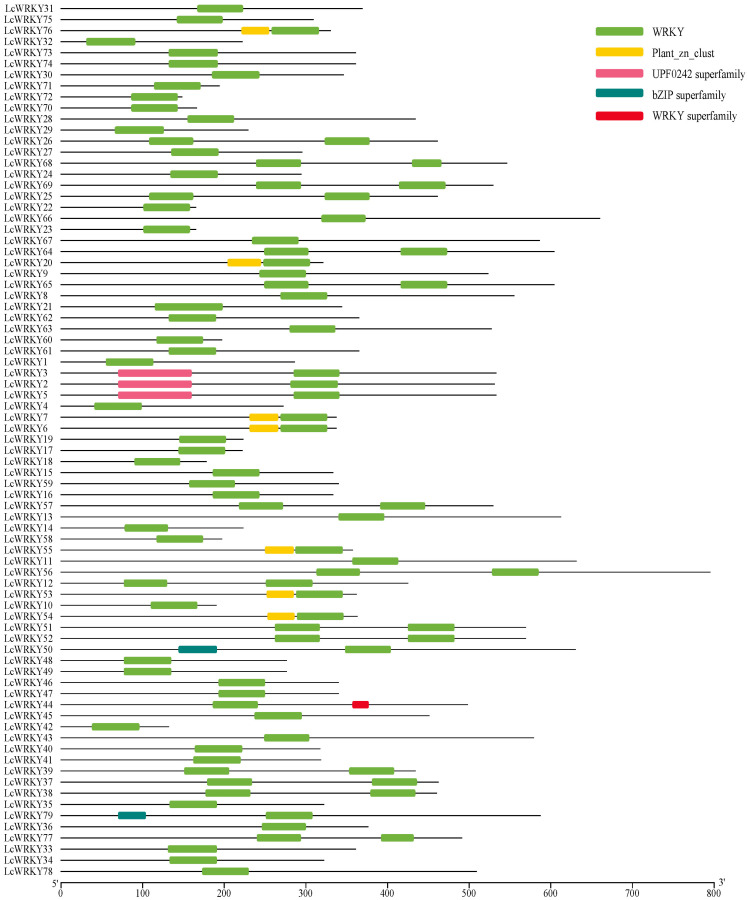
Protein domains of the 79 LcWRKY protein sequences. Different colored blocks represent different protein domains.

**Figure 2 plants-12-02131-f002:**
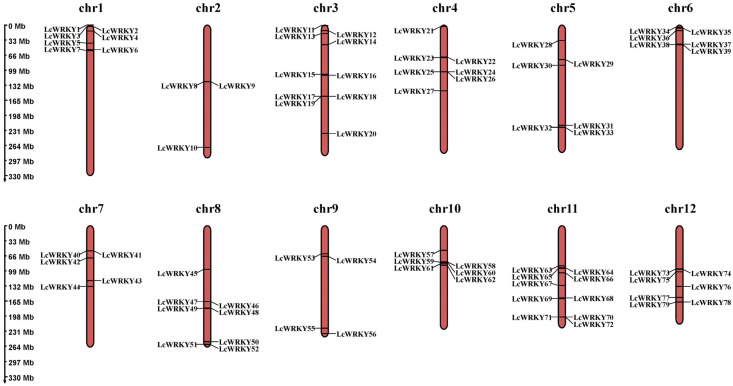
Chromosomal distribution of *LcWRKY* genes. The left scale determines the position of each *LcWRKY* gene on the chromosome.

**Figure 3 plants-12-02131-f003:**
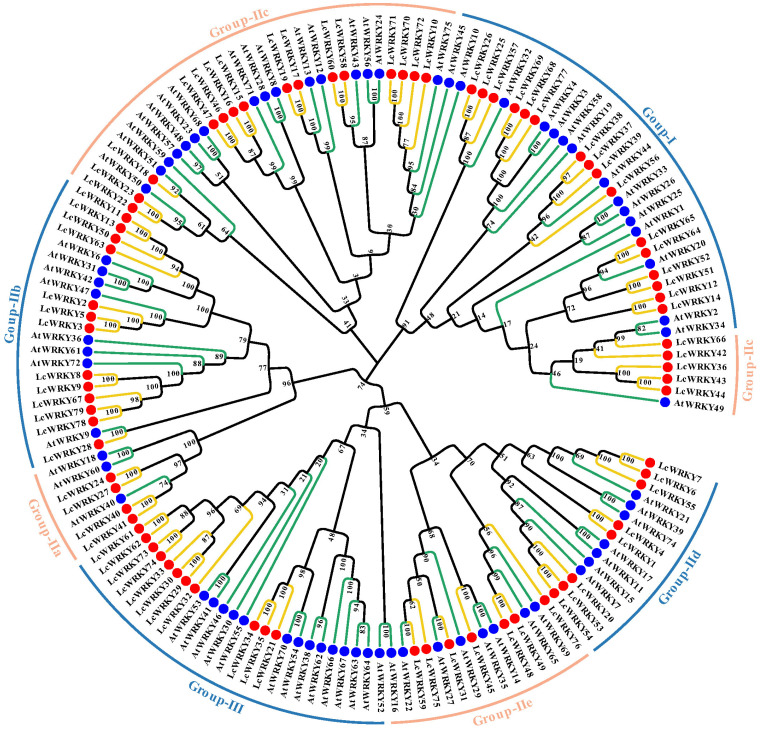
Phylogenetic tree constructed based on the full-length sequences of 72 Arabidopsis WRKYs and 79 LcWRKYs. MEGA11 was used for multi-sequence alignment, and the neighbor-joining (NJ) method was used to construct the phylogenetic tree. To ensure that the structure was credible, the predicted tree was tested using bootstrapping with 1000 replicates. Blue dots and green lines represent Arabidopsis WRKY members, while red dots and yellow lines indicate *L. chinense* var. *rubrum* WRKY members. Different sub-groups are distinguished according to different colored lines.

**Figure 4 plants-12-02131-f004:**
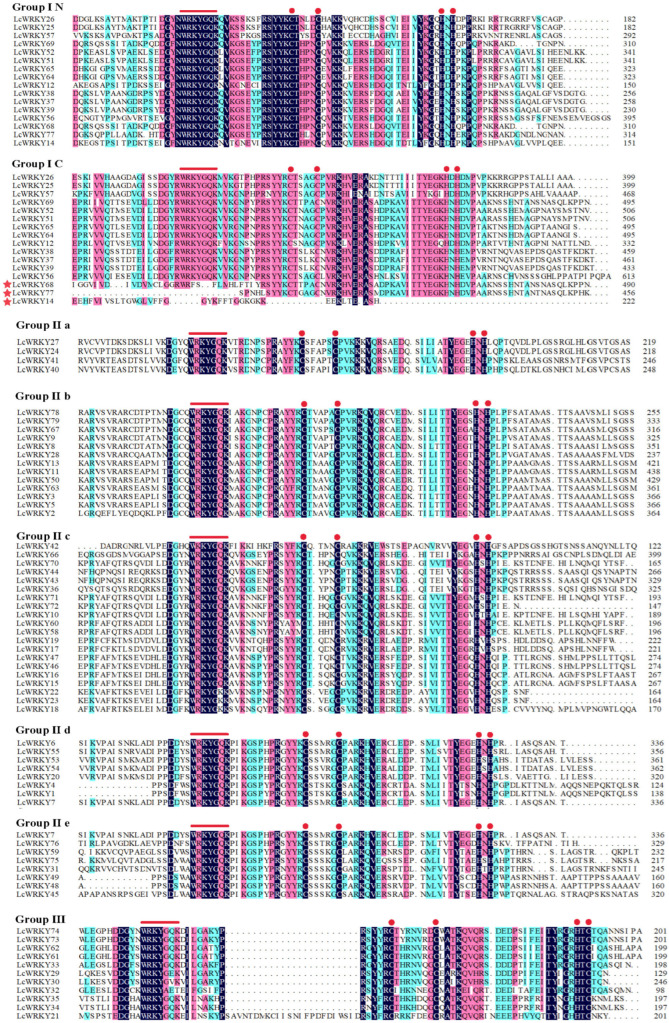
Multiple sequence alignment results for the WRKY domains from LcWRKY members. Alignment was performed using DNAMAN. Different patches of color represent different degrees of similarity in sequences. The red lines represent the WRKY domain, and the red dots represent the zinc-finger domain. A red asterisk indicates special LcWRKYs.

**Figure 5 plants-12-02131-f005:**
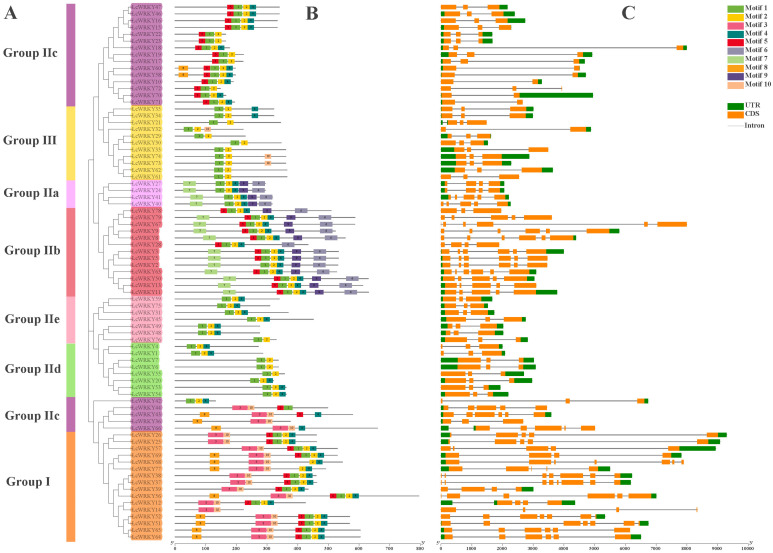
Phylogenetic relationships, conserved motif locations, and gene structures of the *LcWRKYs*. (**A**) An ML tree was constructed using MEGA7 to present the protein sequences of all *LcWRKY* genes. Different colors represent different sub-groups. (**B**) Motif location distribution of the 79 LcWRKY proteins and (**C**) gene structure profile of *LcWRKY* genes.

**Figure 6 plants-12-02131-f006:**
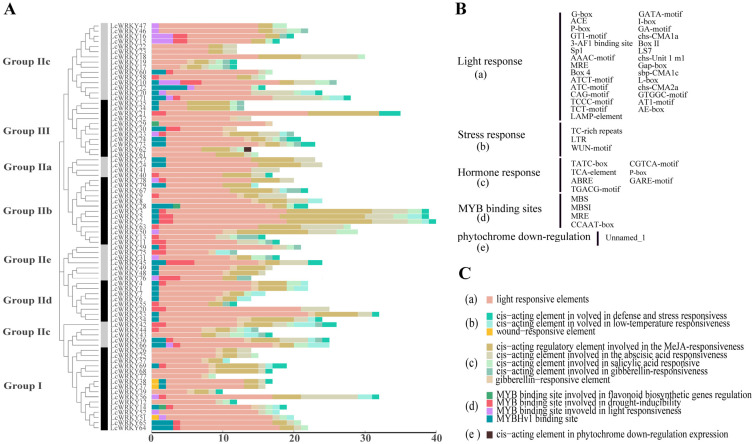
The number of *cis*-acting elements in the upstream 2000-bp region with respect to each *LcWRKY* gene. (**A**) The number of *cis*-acting elements in each *LcWRKY* gene promoter sequence; (**B**) *cis*-acting element abbreviations; and (**C**) explanation of the functional prediction for the *cis*-acting elements.

**Figure 7 plants-12-02131-f007:**
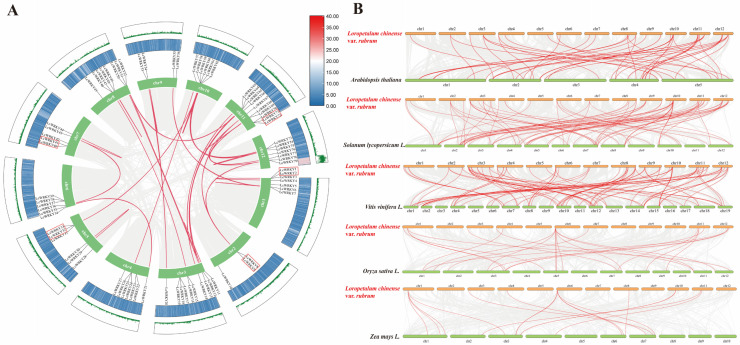
Gene duplication events of *LcWRKYs* and synteny analysis of WRKYs between *L. chinense* var. *rubrum* and other plant species. (**A**) The inter-chromosomal relationships of *LcWRKYs.* The green lines and heat map represent the density of genes on chromosomes. The red box indicates tandem duplicated gene pairs. Duplicate *WRKY* gene pairs are linked by red lines, while the gray lines indicates synteny blocks in the *L. chinense* var. *rubrum* genome. (**B**) Collinear genes are shown with gray lines, while *WRKY* genes are marked with red lines.

**Figure 8 plants-12-02131-f008:**
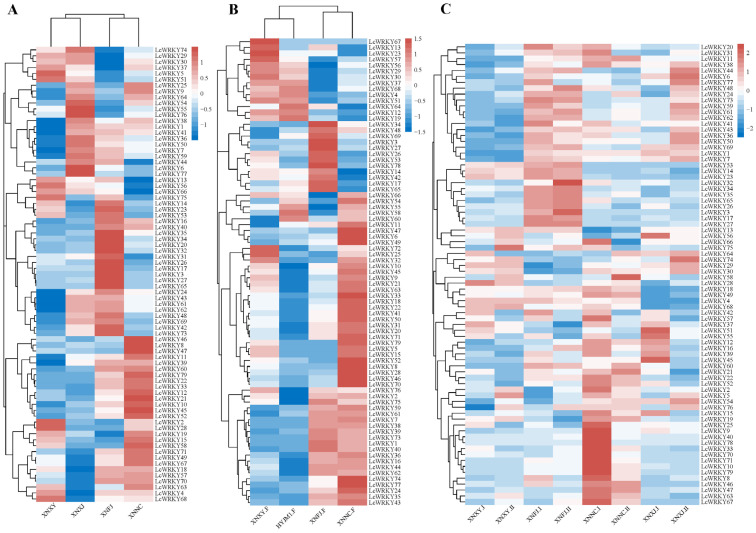
Clustered heat map of *LcWRKY* gene expression patterns, with homogenized rows. The color is indicated from blue to white to red, with the expression ranging from low to high, respectively. (**A**) Expression profile of *LcWRKYs* in mature leaves of four different varieties; (**B**) ‘F’ refers to flower. Expression profile of *LcWRKYs* in flowers of four different varieties; (**C**) ‘Ⅰ’ represents young leaves, while ‘Ⅱ’ represents mature leaves.

**Figure 9 plants-12-02131-f009:**
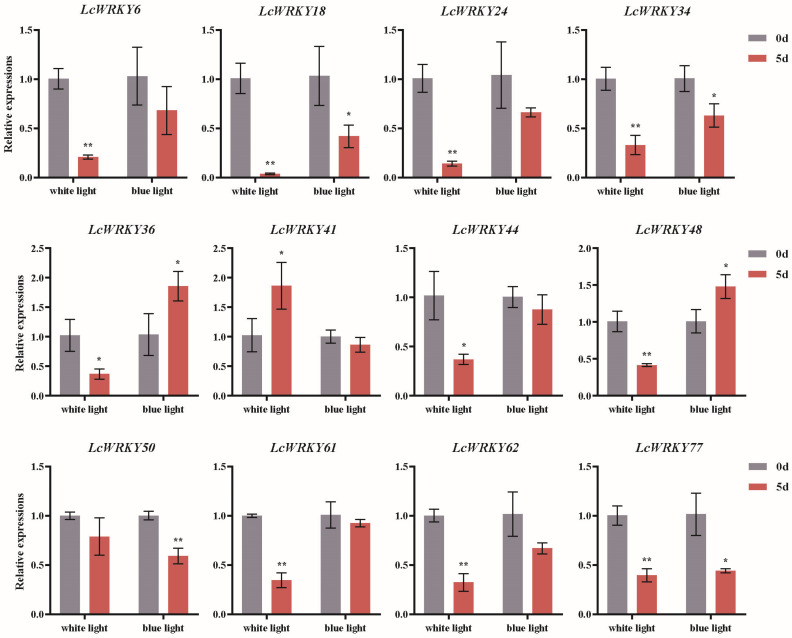
Differential transcription of *LcWRKY* genes in *L. chinense* var. *rubrum* leaves under white and blue light treatments. Different colors represent different treatment times (day 0 vs. 5). Asterisks represent significant differences (*, *p* < 0.05; ***, p* < 0.01).

## Data Availability

Data are contained within the article and Appendix A.

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
