# Peer review of "Genome-Wide Identification and Characterization of WRKY Transcription Factors and Their Expression Profile in Loropetalum chinense var. rubrum"

_plants, 2023, doi:10.3390/plants12112131_

Round 1

Reviewer 1 Report

This is a review of the manuscript titled by "Genome-Wide Identification and characterization of WRKY transcription factors and their Expression profile of Loropetalum chinense var. rubrum".

This study is to identify WRKY TF of <i>Loropetalum chinense</i> var. <i>rubrum</i> and characterize various aspects of the gene family. Using expression profiles of this species' various lines authors investigated expression patterns accordingly.

We have seen many similar styles of studies on the characterization of gene families of interest in various plant species, which is basically descriptive and possibly verbose without concentration on specific topics. The writing structure and tone of this manuscript are reasonable but not focused. I hope that the authors can make possibly emphases on some points in this study. 

Please see my comments and suggestions below. I hope these can help the manuscript be improved.

  1. Is genome sequence data generated from this study? Otherwise, could you specify what genomic sequence data authors used?  You just say 'In the genome of <i>L. chinense</i> var. rubrum, .. ' (as in line 123) but I could not find any information on the genome data or database except reference #44.

Figure 3:

  1. Could you present bootstrap scores on branches? Are they all highly supportive? 
  2. What do stand for red star symbols?
  3. Could you specify what color connect to what species?
  4. I am wondering if there is any association or pattern between promoter region profile and protein subgroups. If any you can discuss with more details. So, it might be interesting to arrange proteins in terms of promoter patterns with subgrouping index (color, etc.) in Figure 6.
  5. Line 262-3: It is an interesting question but you can raise this in the discussion section. It could be reflecting the phylogenetic distance.
  6. I think expression data (with transcriptome analysis) is not consistent with all varieties. I wonder how data were generated with samples grown in controlled conditions. How do you normalize the data? Without proper normalization and avoidance of possible noise due to growing conditions this data is not usable in this manuscript.
  7. In addition, please provide the details on your transcriptome data generation and analysis.

Please see some editorial suggestions below.

  1. Line58: fat-ua > fatua, Petroseli-num > Petroselinum
  2. Lines 91, 99: what is 'etc'? 'et al.'?
  3. Line 107: variant > variety
  4. Line 209: I guess you will have the same result with MEGA11. Please check the version of the program.
  5. Line 520: also, > Also,

Author Response

Dear Reviewer,

Thank you for your detailed review of our manuscript entitled “Genome-Wide Identification and Characterization of WRKY transcription factors and their Expression profile of Loropetalum chinense var. rubrum.” (2343284). The comments are of great help to improving the manuscript. We have studied the comments carefully and perform corresponding corrections in the revised manuscript. The point-by-point responses to the comments and suggestions are listed below.

This is a review of the manuscript titled by "Genome-Wide Identification and characterization of WRKY transcription factors and their Expression profile of Loropetalum chinense var. rubrum". This study is to identify WRKY TF of  Loropetalum chinense var. rubrum and characterize various aspects of the gene family. Using expression profiles of this species' various lines authors investigated expression patterns accordingly. We have seen many similar styles of studies on the characterization of gene families of interest in various plant species, which is basically descriptive and possibly verbose without concentration on specific topics. The writing structure and tone of this manuscript are reasonable but not focused. I hope that the authors can make possibly emphases on some points in this study.

Please see my comments and suggestions below. I hope these can help the manuscript be improved.

  1. Is genome sequence data generated from this study? Otherwise, could you specify what genomic sequence data authors used? You just say 'In the genome of <i>L. chinense var. rubrum, .. ' (as in line 123) but I could not find any information on the genome data or database except reference #44.

Figure 3:

  1. Could you present bootstrap scores on branches? Are they all highly supportive?
  2. What do stand for red star symbols?
  3. Could you specify what color connect to what species?
  4. I am wondering if there is any association or pattern between promoter region profile and protein subgroups. If any you can discuss with more details. So, it might be interesting to arrange proteins in terms of promoter patterns with subgrouping index (color, etc.) in Figure 6.
  5. Line 262-3: It is an interesting question but you can raise this in the discussion section. It could be reflecting the phylogenetic distance.
  6. I think expression data (with transcriptome analysis) is not consistent with all varieties. I wonder how data were generated with samples grown in controlled conditions. How do you normalize the data? Without proper normalization and avoidance of possible noise due to growing conditions this data is not usable in this manuscript.
  7. In addition, please provide the details on your transcriptome data generation and analysis.

Please see some editorial suggestions below.

  1. Line58: fat-ua > fatua, Petroseli-num > Petroselinum
  2. Lines 91, 99: what is 'etc'? 'et al.'?
  3. Line 107: variant > variety
  4. Line 209: I guess you will have the same result with MEGA11. Please check the version of the program.
  5. Line 520: also, > Also,

Q1:Is genome sequence data generated from this study? Otherwise, could you specify what genomic sequence data authors used?  You just say 'In the genome of <i>L. chinense</i> var. rubrum, .. ' (as in line 123) but I could not find any information on the genome data or database except reference #44.

Response: Thanks for the above suggestion. In the revised manuscript, based on your suggestion, we reversed our Materials and Methods :(line 480-line 482) ‘The genomic data of Loropetalum chinense var. rubrum’ were provided by the 'Loropetalum chinense var. rubrum Research Team’ of the College of Horticulture of Hunan Agricultural University..

At present, it is not convenient for the research group to disclose genomic data. If you are interested, please contact the corresponding author: liyanlin@hunau.edu.cn (Yanlin Li).

Q2: Figure 3, Could you present bootstrap scores on branches? Are they all highly supportive?

Response: Thanks for the above suggestion. In the revised manuscript, based on your suggestion, we reversed our Figure and annotation of Figure as below: We modified Figure 3 to present the bootstrap scores. Some of the bootstrap scores are less than 70, which is also found in another similar study, but most of the bootstrap scores are greater than 70.

Q3: What do stand for red star symbols?

Response: Thanks for the above suggestion. In the revised manuscript, based on your suggestion, we reversed our Figure as below: The red star represents two special LcWRKY members in Group 1. To avoid ambiguity we remove red star symbols in Figure 3.

Q4: Could you specify what color connect to what species?

Response: Thanks for the above suggestion. In the revised manuscript, based on your suggestion, we reversed our annotation of Figure as below: We have modified the annotation of Figure 3. (line 198-line 199) ‘Blue dots and green lines represent Arabidopsis WRKY members, while red dots and yellow lines indicate L. chinense var. rubrum WRKY members’

Figure 3. The construction of the phylogenetic tree relies on the full-length sequence of 72 Arabidopsis WRKYs and 79 LcWRKYs, MEGA11 is used for multi-sequence alignment, and the NJ (neighbor-joining) method is used to construct the phylogenetic tree. The predicted tree was tested using bootstrapping with 1000 replicates to make the structure credible. Blue dots and green lines represent Arabidopsis WRKY members, while red dots and yellow lines indicate L. chinense var. rubrum WRKY members. Different subgroups are distinguished according to different colored lines.

Q5: I am wondering if there is any association or pattern between promoter region profile and protein subgroups. If any you can discuss with more details. So, it might be interesting to arrange proteins in terms of promoter patterns with subgrouping index (color, etc.) in Figure 6.

Response: Thanks for the above suggestion. In the revised manuscript, based on your suggestion, we reversed our Figure as below: We reworked Figure 6 to add the classification information of different subgroups of LcWRKY genes. However, the promoters were not significantly associated with different subgroups of LcWRKY genes. However, it does make sense.

Figure 6. The number of cis-acting elements in the upstream 2000bp region of each LcWRKY gene. (A) The number of cis-acting elements in each LcWRKY gene promoter sequence. (B) cis-acting element abbreviation. (C) Explanation of the function prediction of the cis-acting element.

Q6: Line 262-3: It is an interesting question but you can raise this in the discussion section. It could be reflecting the phylogenetic distance.

Response: Thanks for the above suggestion. In the revised manuscript, based on your suggestion, we reversed our discussion as below: (line 434-line 441) ‘Interestingly, the number of gene homologous to LcWRKY genes among dicotyle-donous plants (Arabidopsis thaliana, Solanum lycopersicum L., Vitis vinifera L.) is typically larger than that in monocotyledonous plants (Oryza sativa L., Zea mays L.). This indicates that the closer the relatives relationship between spe-cies, the greater the homology of the WRKY gene family. It can be conjectured that this may be due to differences in plant expansion and evolution caused by the WRKY family of transcription factors.’

Q7: I think expression data (with transcriptome analysis) is not consistent with all varieties. I wonder how data were generated with samples grown in controlled conditions. How do you normalize the data? Without proper normalization and avoidance of possible noise due to growing conditions this data is not usable in this manuscript.

Response: Thanks for the above suggestion. In the revised manuscript, based on your suggestion, ambiguity caused by our writing, we reversed our Materials and Methods as below: (line 551- line 558) ‘Plant materials used for transcriptome sequencing include Loropetalum chinense ‘XNXY’('Xiangnong Xiangyun’) and four L. chinense var. rubrum ‘XNXJ’ ('’Xiangnong Xiaojiao), ‘XNFJ’ ('Xiangnong Fenjiao’), ‘XNNC’ ('Xiangnong Nichang’) and ‘HYJM1’ ('Huaye Jimu 1’) were planted in the flower center of Hunan Agriculrual University.’ and (line 559-line 565 )‘The transcriptome row data reported in this paper were deposited in the Genome Sequence Archive (Genomics, Proteomics & Bioinformatics 2021) in the National Genomics Data Center (Nucleic Acids Res 2022), as well as China National Center for Bioinformation of the Beijing Institute of Genomics, Chinese Academy of Sciences (GSA: CRA009284 and CRA009285) and are publicly accessible at https://ngdc.cncb.ac.cn/gsa. All transcriptome data are expressed using FPKM (Reads Per Kilobase of exon model per Million mapped reads).’

Q8: In addition, please provide the details on your transcriptome data generation and analysis.

Response: Thanks for the above suggestion. In the revised manuscript, based on your suggestion, we reversed our Materials and Methods as below: (line 551- line 558) ‘Plant materials used for transcriptome sequencing include Loropetalum chinense ‘XNXY’ ('Xiangnong Xiangyun’) and four L. chinense var. rubrum ‘XNXJ’('’Xiangnong Xiaojiao), ‘XNFJ’ ('Xiangnong Fenjiao’), ‘XNNC’ ('Xiangnong Ni-chang’) and ‘HYJM1’ ('Huaye Jimu 1’) were planted in the flower center of Hunan Agricultural University. RNA Sample of leaves and flowers preparation and library con-struction and sequencing referring to the previous study [77,78]. Estimation of reads per transcript by expected fragment number per thousand transcript sequences per million sequenced base pairs (FPKM). ’

Q9: Line58: fat-ua > fatua, Petroseli-num > Petroselinum

Response: Thanks for the above suggestion. In the revised manuscript, based on your suggestion, we have changed “fat-ua” to “fatua” and “Petroseli-num” to “Petroselinum” in line 61.

Q10: Lines 91, 99: what is 'etc'? 'et al.'?

Response: Thanks for the above suggestion. In the revised manuscript, based on your suggestion, we have changed “etc” to “et. al” in lines 101, 111.

Q11: Line 107: variant > variety

Response: Thanks for the above suggestion. In the revised manuscript, based on your suggestion, we have changed “variant” to “variety” in line 121.

Q12: Line 209: I guess you will have the same result with MEGA11. Please check the version of the program.

Response: Thanks for the above suggestion. In the revised manuscript, based on your suggestion, our answers to the above questions are as follows: We have tried to use MEGA11 for building ML evolutionary trees, but the software would flash back, so we have used different versions of MEGA software and only MEGA7 is applicable.

Q13: Line 520: also, > Also,

Response: Thanks for the above suggestion. In the revised manuscript, based on your suggestion, we have changed “also” to “Also” in 604.

Once again, thank you very much for your comments and suggestions. A revised manuscript is attached. Should you have any questions, please contact us without any hesitation.

Sincerely yours,

Yanlin Li

23th May, 2023

Reviewer 2 Report

 Comments and Suggestions for Authors

Liu et al. report their work under title “Genome-Wide Identification and characterization of WRKY transcription factors and their Expression profile of Loropetalum chinense var. rubrum”. In this manuscript authors have used bioinformatic tools to characterize WRKY genes and studied their expression profile. I see some aspects which should be improved before the manuscript is accepted for publication.

Major comments

1] Mention “Gene accession” number of all LcWRKY gene.

2] In section 2.8, please study the effect of light on LcWRKY gene expression by RT-PCR and modify the manuscript accordingly.

3] Please edit the manuscript with language editing agency

Minor comments:

Table S4, The term “gene covalency pair” has been mentioned. Please explain it?

Line 71, what is full from of BR?

Line 91, “etc” should be replaced by et. al

Line 507,  what is the meaning of ‘pre-sequencing transcriptome’?

Line 35-“Homoonal response elements”.  What is meant by “Homoonal”?

Author Response

Dear Reviewer,

Thank you for your detailed review of our manuscript entitled “Genome-Wide Identification and characterization of WRKY transcription factors and their Expression profile of Loropetalum chinense var. rubrum.” (2343284). The comments are of great help to improving the manuscript. We have studied the comments carefully and perform corresponding corrections in the revised manuscript. The point-by-point responses to the comments and suggestions are listed below.

Comments and Suggestions for Authors

Liu et al. report their work under title “Genome-Wide Identification and characterization of WRKY transcription factors and their Expression profile of Loropetalum chinense var. rubrum”. In this manuscript authors have used bioinformatic tools to characterize WRKY genes and studied their expression profile. I see some aspects which should be improved before the manuscript is accepted for publication.

Major comments

  1. Mention “Gene accession” number of all LcWRKY gene.
  2. In section 2.8, please study the effect of light on LcWRKY gene expression by RT-PCR and modify the manuscript accordingly.
  3. Please edit the manuscript with language editing agency

Minor comments:

  1. Table S4, The term “gene covalency pair” has been mentioned. Please explain it?
  2. Line 71, what is full from of BR?
  3. Line 91, “etc” should be replaced by et. al
  4. Line 507,  what is the meaning of ‘pre-sequencing transcriptome’?
  5. Line 35-“Homoonal response elements”.  What is meant by “Homoonal”?

Q1: Mention “Gene accession” number of all LcWRKY gene.

Response: Thanks for the above suggestion. In the revised manuscript, based on your suggestion, we uploaded our genes at NCBI and reversed our manuscript as below: (line 129) ‘The NCBI Gene accession number of 79 LcWRKY genes are in Table S5’

Q2: In section 2.8, please study the effect of light on LcWRKY gene expression by RT-PCR and modify the manuscript accordingly.

Thanks for the above suggestion. In the revised manuscript, based on your suggestion, we reversed our manuscript as below:

(line 44-line 48) ‘White light treatment led to a significant decrease in the expression of LcWRKY6, 18, 24, 34, 36, 44, 48, 61, 62, 77 and a significant increase in the expression of LcWRKY41, while blue light treatment led to a significant decrease in the expression of LcWRKY18, 34, 50, 77 and a sig-nificant increase in the expression of LcWRKY36 and 48.’;

(line 341- line 358) ‘Based on the results of the previous gene classification and cis-acting element predic-tions, we selected LcWRKY members distributed in different sub-groups, including LcWRKY6, LcWRKY18, LcWRKY24, LcWRKY34, LcWRKY36, LcWRKY41, LcWRKY44, LcWRKY48, LcWRKY50, LcWRKY61, LcWRKY62 , and LcWRKY77, for RT-PCR analysis under white and blue light treatment. The RT-PCR results indicated that the relative expression levels of LcWRKY6, 18, 24, 34, 36, 44, 48, 50, 61, 62, and 77 were lower after 5 days of white light treatment was than at day 0. Among these, the relative expression levels of LcWRKY6, 18, 24, 34, 48, 61, 62, and 77 were extremely significantly lower on day 5 than on day 0 after white light treatment, while the relative expression of LcWRKY44 was sig-nificantly lower than that on day 0. Only the relative expression of LcWRKY41 was sig-nificantly higher than that on day 0. After 5 days of blue light treatment, the relative expression levels of LcWRKY18, 34, and 77 were significantly decreased, compared to day 0, while the relative expression level of LcWRKY50 was extremely significantly lower than on day 0. However, the relative expression levels of LcWRKY36 and 48 was were significantly higher than on day 0 at after 5 days of blue light treatment. Overall, both white and blue light led to a significant decreases in the relative ex-pression of LcWRKY18, 34, and 77.’;

(line 360- line 362) ‘Figure 9. Differential transcription of LcWRKY genes in L. chinense var. rubrum leaves under white and blue light treatments. Different colors represent different treatment times (day 0 vs. 5). As-terisks represent significant differences  (*, p < 0.05; **, p < 0.01).’;

(line 466-line 471) ‘In this report, we analyzed the expression levels of some LcWRKY genes under while and blue light treatments. Interestingly, both white light and blue light led to a de-crease in the expression of some LcWRKY genes, and whether genes involved in cer-tain photoresponse pathways inhibit their expression and affect certain physi-ological programs in plants need to be further explored for further investigation re-garding the function of LcWRKY genes.’;

(line 568-line 592)

‘4.7. Plant Material and Treatments

The plant material was from the L. chinense var. rubrum variety ‘Hei Zhenzhu’, which is commonly used in landscaping, and yields triennial seedlings. The cultivation substrate was pastoral soil, vermiculite, and perlite in a 3:1:1 ratio. The white and blue light quality settings were as follows: Light quality source selection, custom LED lamp; white light, 390-–780 nm ; blue light, 460 ± 5 nm; and light intensity, about 200 μmol/m2/s . The light source was installed at a height of 15 cm above the plant. A total of 30 pots of plant material were used per treatment. The photoperiod was 14h/10h (day/night), at a temperature of 24 °C/20 °C (day/night) and humidity of 65%–75%. The experiment was conducted at the flower base center of Hunan Agricultural University.

4.8. Real-time fluorescence quantitative PCR

Total RNA was extracted from plant samples using a SteadPure Plant RNA Extraction Kit (Accurate Biotechnology, Hunan), according to the manufacturer’s instructions. cDNA was synthesized in 500 ng total RNA using an Evo M-MLV reverse transcription kit (Accurate Biotechnology, Hunan). Primer design of genes was carried out using the online website https://www.genscript.com/tools/real-time-pcr-taqman-primer-design-tool (see Table S6). The system and procedure of RT-PCR refer to Zhang et. al. [81]. In brief, the total RT-PCR system consisted of 10 µL including 5 µL of 2X SYBR Green Pro Taq HS Premix*, 1 µL of cDNA, 0.8 µL each of upstream and downstream primers (10 µmol/L), and ddH2O to make up to 10µL. The RT-PCR reaction procedure was as follows: Step 1, 95 °C for 30 seconds, ; step 2, 95 °C for 5 seconds, 60 °C for 30 seconds, 72 °C for 10 minutes for 40 cycles, 65 °C for 5 seconds, 95 °C for 5 seconds; 3 repetitions. The relative expression of genes was calculated by the 2-ΔΔCt method.’

(line 602-line 605) ‘Also, the expression levels of some LcWRKYs was were differentially affected by under while white and blue light treatments, , includingfor axample, LcWRKY6, 18, 24 , 36, 48, and 50. et. al, itThese results can be used as support a basis for further in in-depth study.’

Q3: Please edit the manuscript with language editing agency

Response: Thanks for the above suggestion. Based on your suggestion, we will edit the manuscript with a language editing agency.

Q4: Table S4, The term “gene covalency pair” has been mentioned. Please explain it?

Response: Thanks for the above suggestion. In the revised manuscript, based on your suggestion, Due to a mistake in our writing, we changed “gene covalency pair”of Table S4 to “collinear gene pairs”.

Q5: Line 71, what is full from of BR?

Response: Thanks for the above suggestion. In the revised manuscript, based on your suggestion, We added the full name of BR(Brassinosteroid) in line 76.

Q6: Line 91, “etc” should be replaced by et. al

Response: Thanks for the above suggestion. In the revised manuscript, based on your suggestion, we have changed “etc” to “et. al” in line 99 

Q7: Line 507,  what is the meaning of ‘pre-sequencing transcriptome’?

Response: Thanks for the above suggestion. In the revised manuscript, based on your suggestion, To avoid ambiguity, we have made modifications. (line 559- line 564) ‘The transcriptome row data reported in this paper were deposited in the Genome Sequence Archive (Genomics, Proteomics & Bioinformatics 2021) in the National Genomics Data Center (Nucleic Acids Res 2022), as well the China National Center for Bioinformation of the Beijing Institute of Genomics, Chinese Academy of Sciences (GSA: CRA009284 and CRA009285), and are publicly accessible at https://ngdc.cncb.ac.cn/gsa [79].’

Q8: Line 35-“Homoonal response elements”.  What is meant by “Homoonal”?

Response: Thanks for the above suggestion. In the revised manuscript, based on your suggestion, we have changed “Homoonal response elements” to “Hormone response elements” in line 37.

Once again, thank you very much for your comments and suggestions. A revised manuscript is attached. Should you have any questions, please contact us without any hesitation.

Sincerely yours,

Yanlin Li

23th May, 2023

Round 2

Reviewer 1 Report

Most of the comments were properly addressed in the revised version of the manuscript. 

Line 480: The genomic data >> The genome data

Could you add a line below after the statement of data disclosure (Line 480-482) like "Sequencing data used in this study are available upon request to the corresponding author"?

 Line 569: row >> raw

Author Response

Dear Reviewer,

Thank you for your detailed review of our manuscript entitled “Genome-Wide Identification and Characterization of WRKY transcription factors and their Expression profile of Loropetalum chinense var. rubrum.” (2343284). The comments are of great help to improving the manuscript. We have studied the comments carefully and performed corresponding corrections in the revised manuscript. The point-by-point responses to the comments and suggestions are listed below.

Most of the comments were properly addressed in the revised version of the manuscript.

Suggestions for Authors

  1. Line 480: The genomic data >> The genome data

  1. Could you add a line below after the statement of data disclosure (Line 480-482) like "Sequencing data used in this study are available upon request to the corresponding author"?

  1. Line 569: row >> raw

Q1: Line 480: The genomic data >> The genome data

Response: Thanks for the above suggestion. In the revised manuscript, based on your suggestion, we have changed “The genomic data” to “The genome data” in line 437.

Q2: Could you add a line below after the statement of data disclosure (Line 480-482) like "Sequencing data used in this study are available upon request to the corresponding author"?

Response: Thanks for the above suggestion. In the revised manuscript, based on your suggestion, we reversed our manuscript as below: (line 439- line 440) ‘Sequencing data used in this study are available upon request to the corresponding author.’

Q3: Line 569: row >> raw

Response: Thanks for the above suggestion. In the revised manuscript, based on your suggestion, we have changed “row” to “raw” in line 522.

Once again, thank you very much for your comments and suggestions. A revised manuscript is attached. Should you have any questions, please contact us without any hesitation.

Sincerely yours,

Yanlin Li

25th May 2023

Reviewer 2 Report

The revised manuscript still has some minor mistakes:

1] The word “covalency pair” of Table S4 still exist, not replaced by “collinear gene pairs” as commented in response..

2] Line 80, the word overwhelm” is not suitable. Regarding this Comment[8] has not been responded well.

3] Please provide clean version of the revised manuscript, then it will be acceptable for publication.

Author Response

Dear Reviewer,

Thank you for your detailed review of our manuscript entitled “Genome-Wide Identification and Characterization of WRKY transcription factors and their Expression profile of Loropetalum chinense var. rubrum.” (2343284). The comments are of great help to improving the manuscript. We have studied the comments carefully and performed corresponding corrections in the revised manuscript. The point-by-point responses to the comments and suggestions are listed below.

The revised manuscript still has some minor mistakes:

  1. The word “covalency pair” of Table S4 still exist, not replaced by “collinear gene pairs” as commented in response.

  1. In Line 80, the word “overwhelm” is not suitable. Regarding this Comment[8] has not been responded to well.

  1. Please provide a clean version of the revised manuscript, then it will be acceptable for publication.

Q1: The word “covalency pair” of Table S4 still exist, not replaced by “collinear gene pairs” as commented in response.

Response: Thanks for the above suggestion. Based on your suggestion in the revised manuscript, we rechecked and modified the ‘covalency pair’ to ‘collinear gene pairs’ in Table S5.

Q2: In line 80, the word “overwhelm” is not suitable. Regarding this Comment[8] has not been responded to well.

Response: Thanks for the above suggestion. In the revised manuscript, based on your suggestion, we have changed “overwhelm” to “lodging” in line 75.

Q3: Please provide a clean version of the revised manuscript, then it will be acceptable for publication.

Response: Thanks for the above suggestion. Based on your suggestion, we will revise our manuscript and submit a clean version of the revised manuscript.

Once again, thank you very much for your comments and suggestions. A revised manuscript is attached. Should you have any questions, please contact us without any hesitation.

Sincerely yours,

Yanlin Li

25th May 2023
